# Landscape Pattern Change and Ecological Effect in a Typical Mountain–Oasis–Desert Region in the Northwest Region of China

**DOI:** 10.3390/ijerph20054211

**Published:** 2023-02-27

**Authors:** Haiyan Meng, Yi Hu, Zuoji Dong

**Affiliations:** 1School of Economics and Management, University of Chinese Academy of Sciences, Beijing 100190, China; 2National Land Science Research Center, University of Chinese Academy of Sciences, Beijing 100190, China

**Keywords:** ecological restoration, land use and cover change, landscape variation, habitat quality, carbon dynamics, water yield

## Abstract

China has experienced dramatic changes in its land use and landscape pattern in the past few decades. At present, a large number of studies have carried out in-depth and systematic analyses on the landscape variation and its ecological effects in Central and Eastern China, but research on the northwest arid region is relatively deficient. In the present study, the city of Hami, which is located in the northwest arid region of China, was selected as the study area to investigate the responses in the habitat quality, water yield and carbon storage to land use and cover change during 2000–2020. We found that (1) during the entire study period (2000–2020), the variation intensity of the first decade (2000–2010) was significantly greater than that of the second decade (2010–2020), and the conversion between desert and grassland played a dominant role in the conversion among these land types. (2) The maximum value of the habitat degradation degree in Hami city increased during the study period, indicating that the habitat presented a trend of degradation. (3) The total carbon storage in Hami city was approximately 11.03 × 10^6^ t, 11.16 × 10^6^ t and 11.17 × 10^6^ t in 2000, 2010 and 2020, respectively, which indicated an increasing trend. (4) According to the calculation, the average water yield and the total water conservation showed a decreasing trend in the study area. The corresponding results will help to formulate protective measures that are conducive to the restoration of ecosystem functions in extremely arid regions.

## 1. Introduction

Increasingly, intensive human activities are changing the surface conditions of the Earth in various ways, the most significant manifestations of which include changes in land use and cover (LUC) and in landscape patterns [1]. The former, i.e., LUC change (LUCC), includes both the changes in land use (agriculture, pasture or plantation, which describes the human use of land) and land cover (such as forest or desert, which describes the biophysical characteristics of the land surface), and is essentially of a multidisciplinary nature, thus attracting scientists from a range of fields, including but not limited to economics, sociology, geography and geographic information science [2,3,4,5]. The latter, i.e., the change in landscape patterns, which denotes the variation in the allocation mode of multiple land use/cover forms on the regional surface, would cause different environmental effects [4,6,7,8]. Thus, both the changes in the LUC and in landscape patterns successfully represent information about land surface changes through explicit and implicit means. In order to grasp the laws of land surface changes, researchers have performed a great deal of detection work on the variation in LUC and landscape patterns [9,10,11,12,13,14,15]. All of these studies provide important theoretical and technical references for a more comprehensive assessment of the regional surface pattern process.

With the gradual deepening of the research on the change in land surface patterns, scientists have found that the ecological and environmental effects of LUCC are becoming increasingly significant. Therefore, they have begun to evaluate the ecological effects of LUCC and landscape pattern changes. Tran et al. analyzed the relationship between LUCC and land surface temperature (LST) during the process of urbanization and found that LST responds to LUCC types in a nonlinear way [16]. Another research study reported that the trends of urban heat island intensity are spatially correlated with regional land use and change patterns [17]. Soil erosion could also be influenced by LUCC, as it has been demonstrated that the disappearance of forest patches and intensification of cultivation practices may increase soil erosion in a small-scale watershed [18]. Corresponding predictions also revealed that the future LUCC will affect the water balance of the Raccoon River watershed [19]. In addition, it was reported that the projected range contractions of narrow-ranged species are primarily driven by anthropogenic land conversions [20]. These studies show that LUCC has influenced the components (soil, vegetation, water, atmosphere, organisms), functions and stability of various ecosystems. Do these impacts also exist in ecosystems in extremely arid regions? This is a problem that needs systematic and in-depth investigation. Unfortunately, the relevant research is relatively lacking at present.

In the implementation process of the above research, various models and approaches, including Moran’s I index [21], logistic regression [22], the integration of the land surface model and regional climate model [23], the variable infiltration capacity model [24], the Carnegie–Ames–Stanford approach [25], the earth system model [26], the integrated valuation of ecosystem services and tradeoffs (InVEST) model, etc., have played important roles. Among these models and approaches, the InVEST model, which is based on production functions that define how changes in an ecosystem’s structure and function are likely to affect the flows and values of ecosystem services across a land- or a seascape [27], has been proven to possess great applicability and performance in various applications [27,28,29,30,31]. Most recently, the applicability of the model in arid areas has been studied. For example, Zhang et al. discussed the ecosystem service trade-offs and identified eco-optimal regions in urban agglomerations in arid regions of China and found that differences in geospatial heterogeneity could be detected for the trade-offs and synergies of ecosystem services [32]. Fu et al. studied the effects of LUCC and climate change on ecosystem services in Altay Prefecture, China, and reported that the model could be reliably used to evaluate the ecosystem services [33]. Wang et al. investigated the effects of the Grain for Green Program on the water ecosystem services in an arid area of China, and the model performed well according to their research [34]. Although these studies have increased our understanding of the applicability of the model in arid areas, a discussion of the applicability of the model in extremely arid areas is still relatively lacking.

As the most populous country in the world, China has experienced dramatic changes in LUC and landscape patterns in the past few decades. Located in the west of China, Xinjiang is the largest province in China. The Hami city region, located in the east of Xinjiang, is an extremely arid region (mean annual precipitation less than 40 mm) with various types of surface coverage, large topographic relief, a relatively fragile ecological environment and great resource endowment potential. It is of great significance in the protection of the ecological environment and the pursuit of sustainable development to accurately study the changes in the land surface pattern in this region (extremely arid region) and their ecological effects. In view of this, the present study aims to (1) identify the characteristics of LUC and landscape pattern changes in Hami city in the past 20 years; (2) analyze the specific response of habitat quality, water conservation and carbon storage to landscape pattern changes in Hami city; and (3) put forward suggestions for an ecological protection strategy in Hami city.

## 2. Materials and Methods

### 2.1. Study Area

Hami city (40°52′–45°05′ N, 91°06′–96°23′ E) is in the east of Xinjiang, approximately 404 km from east to west, and 440 km from north to south, with a total area of 142,100 square kilometers. It is adjacent to Jiuquan City, Gansu Province in the east and southeast; Bayingolin Mongol Autonomous Prefecture in the south; Changji Hui Autonomous Prefecture and Turpan City in the west and southwest; and Mongolia in the north and northeast. It is one of the most important ports through which Xinjiang develops border trade with Mongolia. The topography of Hami city can be summarized as four mountains and three basins, with large topographic differences. The central part is the main vein of the Tianshan Mountains, extending in a northeast–southwest trend. The north and south sides are middle and low mountain areas, which account for three fifths of the total area of the city. Hami city has a temperate continental arid climate, but due to the influence of the Tianshan Mountains, the climate difference is obvious. Generally, the south of the Tianshan Mountains is dry and hot, with relatively little precipitation. The north of the Tianshan Mountains is relatively humid and cool with more rainfall. On average, the city receives 3285 h of bright sunshine annually, making it one of the sunniest nationally. The monthly 24-h average temperature ranges from −9.8 °C in January to 26.8 °C in July, while the annual mean is 10.25 °C (Figure 1).

### 2.2. Dataset and Model Building

#### 2.2.1. Data and Parameters

The data and parameters used in the study include precipitation, annual potential evapotranspiration, land use type, soil depth, root depth, plant available water, runoff coefficient, topographic index, soil saturation hydraulic conductivity, etc. The sources and basic information of the collected data are shown in Table 1. In order to ensure the uniformity of the data input into the model, all the data lists in Table 1 were re-sampled to a resolution of 30 m × 30 m using QGIS software. The original geographical coordinate system of the collected data is WGS 84 and all data were reprojected to CGCS 2000 (projected coordinate system, Datum: China 2000, Ellipsoid: CGCS2000, Prime meridian: Greenwich).

#### 2.2.2. Land Use/Cover Change

The LUC transfer matrix was used to explore the direction and quantity of transfer among LUC types; the matrix can reflect the structural characteristics of the regional LUCC [39]. Using the spatial analysis function of QGIS, the LUC types of the study area in different periods were spatially superimposed to obtain the LUCC from 2000 to 2010, 2010 to 2020 and 2000 to 2020; the matrix is denoted by Formula (1).
(1)$$S_{ij=}\left[\begin{array}{ccc} S_{11} & \cdots & S_{1n}\\ \vdots & \ddots & \vdots \\ S_{n1} & \cdots & S_{nn} \end{array}\right]$$
where *S* represents the area of each LUC type, *i* and *j* are the LUC types at the beginning and end of the study period, and *n* is the number of LUC types.

#### 2.2.3. Landscape Change

Generally, the characterization of the landscape can be performed using landscape indices (Table 2); as a result, the present study uses the variations in landscape indices to investigate the variation in the landscape during the study period.

#### 2.2.4. Habitat Quality Assessment Model

Habitat quality refers to the ability of the ecosystem to provide conditions suitable for the sustainable survival of individuals and populations. It is usually regarded as a continuous variable in the modeling process, and high-quality habitats are relatively complete. Habitat quality depends on the habitat background conditions, the proximity to human land use and the intensity of them. In general, habitat quality will deteriorate with the increase in land use intensity nearby [40,41,42]. The analysis of habitat quality was conducted using the habitat quality model (HQM) in the InVEST model, which combines LUCC and biodiversity threat information to generate a habitat quality map. This approach produces two sets of key information that can help to make an initial assessment of the protection needs: the relative extent and degradation of different types of habitats in an area, and changes over time. This approach further allows the rapid assessment of habitat conditions and changes as a more detailed measure of the biodiversity status. In the model, the degree of habitat degradation can be obtained by calculating the negative impact of the threat source on the habitat, and the suitability of the habitat is combined with the suitability of the habitat to obtain the level of habitat quality. The higher the sensitivity of the habitat to the threat factor, the greater the influence of that factor on the habitat degradation. If the site or habitat type at raster *x* is *y*, the degree of habitat degradation (*D_xy_*) at this point can be expressed as: (2)$$D_{xy}=\overset{R}{\underset{r=1}{\sum }}\overset{Y_{r}}{\underset{y=1}{\sum }}\left(\frac{w_{r}}{\sum_{r=1}^{R}w_{r}}\right)r_{y}i_{rxy}\beta_{x}S_{jr}\;$$
where r is the raster cell, $Y_{r}$ is the total number of raster cells occupied by the threat factor r, $w_{r}$ is the normalized threat weight, $r_{y}$ is used to determine y whether the raster is the source of the threat factor r, $\beta_{x}$ represents the accessibility of the raster to the threat source under social, legal, etc., protection status $S_{jr}$ represents the sensitivity of the land cover type j to the threat factor r, and $i_{rxy}$ is the distance function between the habitat and the threat factor, and is given by the following:(3)$$\mathrm{Linear}:\;\;\;\;i_{rxy}=1-\left(\frac{d_{xy}}{d_{rmax}}\right)$$
(4)$$\mathrm{Exponential}:\;\;\;\;i_{rxy}=\exp\left[-\left(\frac{2.99}{d_{rmax}}(d_{xy}\right)\right]$$

According to the above equation, the habitat quality ($Q_{xj}$) of the land cover type at a site is
(5)$$Q_{xj}=H_{j}\left[1-\left(\frac{D_{xj}^{z}}{D_{xj}^{z}+k^{z}}\right)\right]$$
where $H_{j}$ is the habitat suitability of the LUC type, $D_{xj}$ is the habitat stress level of the raster x cell in the LUC type j, z is a constant and k is the half-saturation factor, with a default value of 0.5.

Referring to the InVEST model and to the guidelines [37], as well as the south handbook and related research [43,44,45], and based on the actual situation of the Hami city region, the cropland (CL), urban construction land (UCL), rural land (RRL), other construction land (OCL) and unused land (UL) are taken as the threat source for habitats; the maximum influence distance and weight of the threat source factors, the suitability of different types of habitats and the sensitivity of each habitat to the threat factors are firstly determined, and a table of habitat quality module parameters is designed (Table 3 and Table 4).

#### 2.2.5. Carbon Storage

The InVEST carbon storage and sequestration model uses maps of the LUC, along with stocks in aboveground biomass ($C_{i\_above}$), belowground biomass ($C_{i\_below}$), soil ($C_{i\_soil}$) and dead organic matter carbon pools ($C_{i\_dead}$), to estimate the amount of carbon stored in a landscape. The InVEST carbon storage and sequestration model aggregates the total carbon in these four pools according to the LUC maps. The model delineates carbon storage densities to the LUC data which may include classes, such as forest, grassland or agricultural land. For each LUC type, the model requires an estimate of the amount of carbon in at least one of the four fundamental pools described above. The model simply applies these estimates to the LUC map to produce a map of carbon storage in the carbon pools included. The average carbon density of the four carbon pools can be calculated using the following formula:(6)$$C_{T}=\overset{n}{\underset{i=1}{\sum }}C_{i\_T}=\overset{n}{\underset{1}{\sum }}A_{i}\times \left(C_{i\_above}+C_{i\_below}+C_{i\_dead}+C_{i\_soil}\right)\;$$
where *C_T_* is the total regional carbon storage, $C_{i\_T}$ is the carbon storage of LUC type i, and *Ai* is the area of LUC type I; $C_{i\_above}$, $C_{i\_below}$, $C_{i\_dead}$, $C_{i\_soil}$ represent carbon storage in the aboveground carbon pool (t/hm^2^), belowground carbon pool (t/hm^2^), soil carbon pool (t/hm^2^) and dead leaf carbon pool (t/hm^2^) [27,38,46]. When estimating the carbon storage, a carbon density table (Table 5) is required in addition to the LUC data. The tables set out in this paper are based on relevant empirical studies in arid zones.

#### 2.2.6. Water Yield

The water yield of the InVEST model assesses the capacity of different regions to provide water for human consumption and further analyzes the impact of the LUCC on this capacity. The water conservation model calculates the water conservation capacity of each grid cell in the study area based on the Budyko theory. The water yield is calculated using the following formula:(7)$$Y_{xj}=\left(1-\frac{AET_{xj}}{P_{x}}\right)\times P_{x}$$
where, $Y_{xj}$ represents the annual water yield (mm) of land cover type j in raster cell x; $AET_{xj}$ represents the annual evapotranspiration (mm) of land cover type j in raster cell x; $P_{x}$ is the annual precipitation (mm) of raster cell x, obtained from the following equation:(8)$$\frac{AET_{xj}}{P_{x}}=\frac{1+\omega_{x}R_{xj}}{1+\omega_{x}R_{xj}+\frac{1}{R_{xj}}}$$
where $R_{xj}$ is the dryness index, which can be calculated by $R_{xj}=\frac{K_{x}E_{T_{0}}}{P_{x}}$, $K_{x}$ is plant (vegetation) evapotranspiration coefficient associated with the LUC and $E_{T_{0}}$ is the reference evapotranspiration from pixel x; $\omega_{x}$ is the ratio of the annual vegetation water requirements to the total precipitation, which can be calculated by $\omega_{x}=Z\frac{AWC_{x}}{P_{x}}$, where *Z* is the Zhang coefficient, a constant that characterizes the seasonal factor, with data fluctuating between 1 and 30; finally, $AWC_{x}$ is the quantity of total plant available water that the soil can provide.

During the calculation of water conservation, a correction was made by combining the soil saturation conductivity, runoff coefficient and topographic index as:(9)$$Retention=min\left(1,\frac{249}{Velocity}\right)\times min\left(1,\frac{0.9\times TI}{3}\right)\times min\left(1,\frac{K_{sat}}{300}\right)\times Yield$$
where retention is the water conservation (mm); velocity is the runoff coefficient; $K_{sat}$ is the soil saturation conductivity (mm/d), calculated by the Neuro Theta based on the clay, silt, and sand content of the soil, TI is the topographic index, which can be calculated by Formulas (2)–(10).
(10)$$TI=\left(\frac{{\mathrm{Drainage}}_{\mathrm{Area}}}{{\mathrm{Soil}}_{\mathrm{Depth}}\times {\mathrm{Percent}}_{\mathrm{Slope}}}\right)$$
where drainage_Area_ represents the number of grids in the catchment area, soil_Depth_ is the soil depth (mm), and percent_Slope_ is the slope in percent [47,48,49].

The main input data required for the water yield model are the LUC data, annual rainfall, annual potential evapotranspiration, soil depth, plant available water, watershed boundary and biophysical parameters. The biophysical coefficients include the evapotranspiration coefficient and root depth (Table 6). The evapotranspiration coefficient and root depth are assigned according to the relevant literature and in conjunction with the FAO.

## 3. Results

### 3.1. LUCC

The unused land and grassland in Hami city accounted for approximately 95% of the total area, while the proportion of cultivated land, forest, construction land and water was relatively low (Figure 2 and Figure 3). From 2000 to 2010, the conversion of various LUC types was mainly concentrated among unused land, grassland, cultivated land and forest. Among them, the conversion area of unused land with other types was relatively large. Specifically, the area of unused land converted to grassland was the largest (approximately 9638.80 km^2^), the area converted to cultivated land was 153.80 km^2^, the area converted to water was 127.69 km^2^, and the area converted to construction land was 66.76 km^2^. The area converted from grassland to unused land was 3163.68 km^2^. The conversion among other types also included 590.31 km^2^ of grassland into cultivated land, 262.16 km^2^ of grassland into forest, 361.99 km^2^ of forest into grassland, 105.27 km^2^ of cultivated land into grassland, etc. From 2010 to 2020, the conversion intensity between unused land and other types was significantly reduced: 194.48 km^2^ to water and 96.8 km^2^ to grassland. In addition, the areas converted from grassland to cultivated land and from cultivated land to grassland were 93.55 km^2^ and 69.74 km^2^, respectively. During the entire study period (2000–2020), the variation intensity of the first decade (2000–2010) was significantly greater than that of the second decade (2010–2020), and the conversion between desert and grassland played a dominant role in the conversion among these land types.

### 3.2. Variations in Landscape Metrics

The landscape metrics of the LUC types are summarized in Table 7. From 2000 to 2020, the NP of CL, FL, GL, WB and CSL increased, whereas that of UL decreased. The LSI of CL, FL, WB and CSL decreased compared with that of GL and UL. The PARA of FS and GL increased, whereas the parameters of other LUC types did not show a clear trend. For ENN, only the GL showed a decreasing trend, as demonstrated in Table 7. 

We also investigated the variation in nine landscape metrics for the entire study area (Figure 4). Overall, the NP, LSI and SHDI increased during the study period. The similar increasing pattern of the NP and SHDI demonstrated an increase in the number of patches, overall fragmentation, regional landscape compactness and heterogenicity in the study area (Figure 4). In contrast, the decrease in the LPI, MPFD and CONTA, together with the decrease-increase in the MSI and ENN, further reflected the obvious trend of mutual penetration and proximity of patches brought about by the rapid LUC change from 2000 to 2010 (Figure 4). Here, the rapid decrease in the MSI is related to the substantial increase in the conversion of UL to GL. At the same time, the contrast between the increase in the NP, SHDI and LSI and the decrease in the PARA and MSI in the landscape from 2010 to 2020 reflects how the shape represented by cultivated land patches tends to be more regular (Figure 4).

The local characteristics of the landscape in Hami city were detected at a resolution of 10 km (Figure 5). The results showed that the spatial distribution characteristics of the SHDI and LSI were consistent. The high values of the SHDI and LSI mainly appeared in the middle of the study area and overlapped with the non-unused land in space. The highest values of the two parameters were spatially overlapped with the ecotone (where CL, GL and UL appeared simultaneously) around the oasis. The spatial distribution of PARA was similar to that of SHDI, but the distribution of its highest value was scattered. The high-value area of ENN was more dispersed; because this parameter’s value is easily affected by the specific location and number of types of grids, its description of ENN for characteristics, such as shape variation and fragmentation are insufficient. The distribution of the CONTAG index is similar to that of SHDI, but it has a wider distribution area. Its high value is often distributed in the mixed grid of UL and other land types, while the distribution of multiple patches in the inner region of the oasis reduces the value range of CONTAG.

### 3.3. Ecological Effect

#### 3.3.1. Habitat Quality

In general, the maximum value of the habitat degradation degree in Hami city increased during the study period, from 2000 to 2020, it was 0.12, 0.13 and 0.14, respectively, indicating that the habitat presented a trend of degradation (Figure 6a–c). In order to present the spatial heterogeneity of the habitat quality more intuitively, the habitat in Hami city was divided into four grades, excellent (0.75–1), good (0.5–0.75), medium (0.25–0.5), and poor (0–0.25), using spatial analysis tools. The number of grids at each level in each year was counted, and the spatial distribution of habitat quality at each level was calculated (Figure 6d–f). The results showed that, in each period, the area with general habitat quality accounted for the highest proportion (more than 78.80%). At the same time, the area proportion of this type showed a downward trend (83.33%, 79.07% and 78.84% in 2000, 2010 and 2020, respectively). The proportion of areas with poor habitat quality increased from 0.13% in 2000 to 0.24% in 2010, and further increased to 0.44% in 2020. The proportion of areas with good habitat quality increased significantly from 2000 to 2010 and decreased slightly from 2010 to 2020. The proportion of good habitat quality decreased from 2010 to 2020 (Figure 6d–f). In order to further obtain the variation in habitat quality in the study area, the spatial analysis tool was used to obtain the habitat quality grid map (Figure 6g–i). The results showed that from 2000 to 2010, the areas with better habitat quality were mainly distributed in Balikun Kazakh Autonomous County, and the areas with worse habitat quality were mainly distributed in Yizhou County. From 2010 to 2020, the areas where habitats varied were mainly concentrated around Yizhou County, and the variations in other areas were not obvious.

#### 3.3.2. Carbon Storage

The results of the carbon storage assessment are dependent on a table of carbon densities corresponding to landscape types. Therefore, this paper focuses on the analysis of the model output data of carbon storage in Hami city. Landscape data and carbon pool density tables were entered into the InVEST model and run to obtain carbon storage assessment results for the study area. At the same time, spatial mapping was carried out with the help of QGIS software to obtain the spatial distribution of carbon storage in the three periods of 2000, 2010 and 2020, respectively (Figure 7). Overall, the average carbon storage and total carbon storage in the study area showed an increasing trend from 2000 to 2020. The total carbon storage in Hami city was approximately 11.03 × 10^6^ t, 11.16 × 10^6^ t and 11.17 × 10^6^ t in 2000, 2010, and 2020, respectively, with a slight increase in 2020. On the scale of individual counties, from 2000 to 2020, Yizhou County had the largest total carbon storage, Barkol Kazak Autonomous County was ranked second, and the smallest was recorded in Yiwu County. Among them, both Barkol Kazak Autonomous County and Yiwu County maintained an increasing trend, while Yizhou County showed a decreasing trend followed by an increasing trend. In terms of spatial distribution, the high-value areas of carbon storage in Hami city were all concentrated in the mountainous and forested areas at high altitudes, which have high vegetation cover and high precipitation, which is conducive to carbon storage. With the decrease in altitude, the carbon storage presents a decreasing trend (Figure 7).

#### 3.3.3. Water Conservation Capacity

The InVEST water conservation model was used to obtain the water supply function of Hami city in 2000, 2010 and 2020. According to the calculation, the average water yield and the total water conservation showed a decreasing trend in the study area from 2000 to 2020 (Figure 8). The average water yield per unit area in the study area ranged from 13.49 mm to 28.78 mm from 2000 to 2020, and the total water conservation ranged from 15.22 × 10^8^ m^3^ to 69.32 × 10^8^ m^3^. In terms of individual counties, the average water yield and total water conservation of Yizhou County, Yiwu County and Barkol Kazak Autonomous County all showed a decreasing trend from 2000 to 2020 (Figure 9). Among these three counties, the average rate of decline in water yield was 1.02 mm/10a, 2.68 mm/10a and 2.05 mm/10a in Yizhou, Yiwu and Barkol, respectively. The total water conservation of the three counties decreased by 10.84 × 10^8^ m^3^, 24.2 × 10^8^ m^3^ and 32.37 × 10^8^ m^3^ respectively. Barkol Kazak Autonomous County showed the greatest decrease in water conservation among the three counties.

## 4. Discussion

The spatial distribution and change in habitat degradation shows that the regions with rapid economic development are more prone to habitat degradation [50]. In this study, the degradation of habitats in some areas appears as a ring structure, increasing from the center to the periphery. In areas (cities) with a high intensity of human activities, the habitat degradation is low. With the increase in the distance from the city, the degree of habitat degradation gradually increases. This is because the habitats in urban areas have been seriously damaged, and the quality of habitats cannot continue to decrease. In the ecotone around the city, urban expansion occupies grasslands, which has a certain negative impact on the original habitat, as has been reported [51]. This study also found that in the oasis desert transition zone, the habitat degradation is high due to both the natural environment and human factors. In areas with low intensity of human activities and great difficulty in land development, the habitat is less affected and basically maintains its original state. In addition, the degree of degradation is related to the size of habitat patches. In other words, the more fragmented the patches, the higher the degree of degradation.

Overall, the high-value carbon storage area in the Hami city region was concentrated in the southwest area of Barkol Kazak Autonomous County, and the carbon storage capacity of Barkol Kazak Autonomous County gradually expanded to the northwest. The main reason was that Barkol Kazak Autonomous County is located in the grassland between the eastern part of the Tianshan Mountains and the East Junggar fault block system, where rainfall is high, and the grassland area of Barkol Kazak Autonomous County increased by 5350.46 km^2^ from 2000 to 2020, resulting in an increase in vegetation cover and enhanced carbon storage capacity. During this process, the warm and wet trend of Northwestern China would have contributed to the increase in carbon sequestration, as has been reported [52]. Due to the complex response of terrestrial carbon storage to climate and LUC variation [53,54], more systematic research is needed to verify the carbon storage of the disturbed terrestrial ecosystem to the future climatic and human activity scenarios. On the county-level, the carbon reserves in Yizhou County gradually increased during the period of 2000–2020. This was mainly due to the geographical location of the region (Yizhou County is in the center of the entire Hami city) and the influence of anthropogenic activities. In addition, we found that forests in Hami city decreased (by 137.43 km^2^) and cropland and construction land increased (by 485.27 km^2^ and 285.19 km^2^, respectively) during the period of 2000–2020, resulting in a weaker carbon storage capacity, which further proves that the carbon storage in regions with strong human activity will decline [55]. The high-value area of carbon storage in Yiwu County was mainly concentrated in the southwest part of the county and it showed an increasing trend from 2000 and 2020, which contributed to the increase in the carbon storages of Hami city.

Most of the study area has a low water yield; comparatively, the high-value areas of water yield were concentrated in the high-altitude mountains in the north of Hami city. Water yield was heavily influenced by precipitation and evaporation [47]. Water yield and precipitation in the study area have a consistent spatial distribution; this is because high-altitude mountainous areas are characterized by high precipitation and low evaporation, thus corresponding to relatively higher water yields, which in turn correlates with a higher water conservation function [56]. As the altitude decreases, the gradual decrease in precipitation and increase in evaporation may contribute to a lower water yield, which restricts the water conservation function of the corresponding ecosystems [57,58]. On the level of individual counties, the southwestern part of Barkol Kazak Autonomous County, Yiwu County, and the northern part of Yizhou County had a higher water yield and water conservation capacity. This was due to the region’s high-altitude mountains, high rainfall, and large areas of grassland and forest, where temperatures and evapotranspiration were low. 

As a typical extremely arid region, the ecological effect of landscape variation in our study area presented different characteristics compared with other regions. Specifically, the habitat degraded, as displayed in Figure 6. The average water yield and the total water conservation also showed a decreasing trend (Figure 8 and Figure 9). In contrast, the total carbon storage was approximately 11.03 × 10^6^ t, 11.16 × 10^6^ t and 11.17 × 10^6^ t in 2000, 2010, and 2020, respectively, and showed an increasing trend during the study period. The degradation of species habitat and decreasing trend of the water conservation capacity was mainly attributed to the conversion of unused land to artificial grasslands, cropland and construction land. Our study is in line with a previous study, which found that habitat degradation in the Winike watershed (Southwest Ethiopia) was continuously increasing over the last three decades and the most contributing factor was agricultural expansion. In addition, population density and land-use intensity might also have contributed [42]. In contrast, the decreasing trend of the water yield and the total water conservation in our study area is different from that of Jing-Jin-Ji, China [59], Haraz Basin, Iran [60] and Mereb-Gash River Basin in the Horn of Africa [61]. The sparse vegetation of our study area might contribute to this difference. As to the carbon storage, the relatively higher carbon concentration of increased artificial grasslands and cropland resulted in the increase of total carbon storage in our study area, which is also different from that of previous studies [62,63,64], which also demonstrated the influence of natural background (arid and sparse vegetation) to the positive response of carbon storage to the LUCC. These unique responses (of habitat quality, carbon storage and water yield to landscape change) indicate that different protection or restoration strategies should be implemented for the various landscapes in Hami city. 

The InVEST model added all threats on the landscape to estimate the loss of habitats, although there is evidence that, in some cases, the collective impact of multiple threats is far greater than the sum of individual threat levels. Since the selected landscape of interest is usually nested in a larger landscape, it must be recognized that the landscape has an artificial boundary—that is, habitat threats outside the research boundary have been ignored. Therefore, at the edge of a given terrain, the threat intensity will always be small [65]. Therefore, we suggest two approaches to minimize the modeling error: firstly, we may select a landscape where the spatial extent obviously exceeds the boundary of the landscape of interest for modeling; secondly, we could limit the analysis to landscapes where the sources of degradation are concentrated in the middle of the landscape.

Using maps of the LUCC and the carbon storage of the corresponding LULC, the InVEST model estimates the net carbon stored in a land parcel over time [66]. Although a number of studies have used the model to estimate the carbon dynamics of the LUCC in terrestrial ecosystems, the model is still limited in simulating carbon storage. Firstly, the variation in carbon storage only depends on the LUCC and the carbon density of a specific LULC type; thus, the variations in carbon density among the different sub-LULC types is neglected [27]. Secondly, the calculation procedure ignores the flux in carbon between terrestrial ecosystems and the atmosphere and soil, which further adds uncertainty to the carbon storage of the LUCC [52]. In summary, the estimation of carbon storage is oversimplified. Biophysical conditions that are important for carbon sequestration, such as photosynthesis rates and the presence of active soil organisms, are also not included in the model.

The model also has two main limitations when modeling the water yield. Firstly, water transferred for irrigation, either between subbasins or between seasons, is not well captured by the model. When applying the empirical approach to cropland, irrigation patterns should be considered [47]. Secondly, the model greatly simplifies the consumption demand. For each LULC, a variable is used to represent multiple aspects of water resource allocation, which may distort the complex distribution of water between different uses and over time [67].

## 5. Conclusions

The objective of this study was to analyze the LUCC and landscape variation and its ecological response of Hami City. The LUC transfer matrix was used to explore the direction and quantity of transfer among LUC types, habitat quality, carbon storage and water yield. These were modeled using the InVEST model, and the results obtained in this study are as follows:(1)From 2000 to 2010, the conversion of various LUC types was mainly concentrated among unused land, grassland, cultivated land and forest. From 2010 to 2020, the conversion intensity between unused land and other types was significantly reduced.(2)In general, the maximum value of the habitat degradation degree in Hami city increased during the study period, from 2000 to 2020, it was 0.12, 0.13 and 0.14, respectively, indicating that the habitat presented a trend of degradation.(3)Overall, the average carbon storage and total carbon storage in the study area showed an increasing trend from 2000 to 2020.(4)The average water yield and the total water conservation showed a decreasing trend in the study area from 2000 to 2020.

## Figures and Tables

**Figure 1 ijerph-20-04211-f001:**
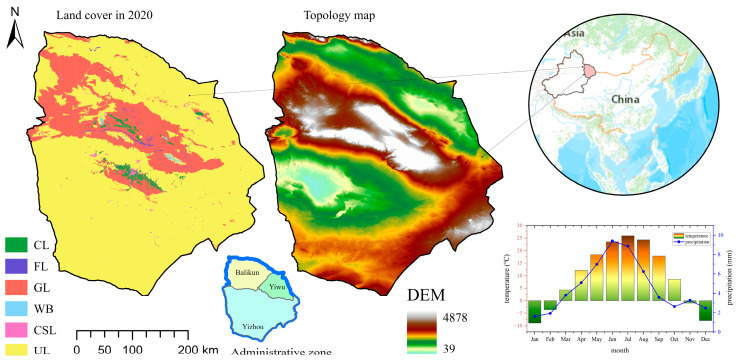
Study area. The left one shows the land use/cover (CL, FL, GL, WB, CSL and UL are cropland, forest, grassland, water body, construction land and unused land, respectively) of the study area, the middle one is the digital elevation model of Hami city, the lower-right graph shows the mean monthly temperature and mean monthly precipitation of Hami city.

**Figure 2 ijerph-20-04211-f002:**
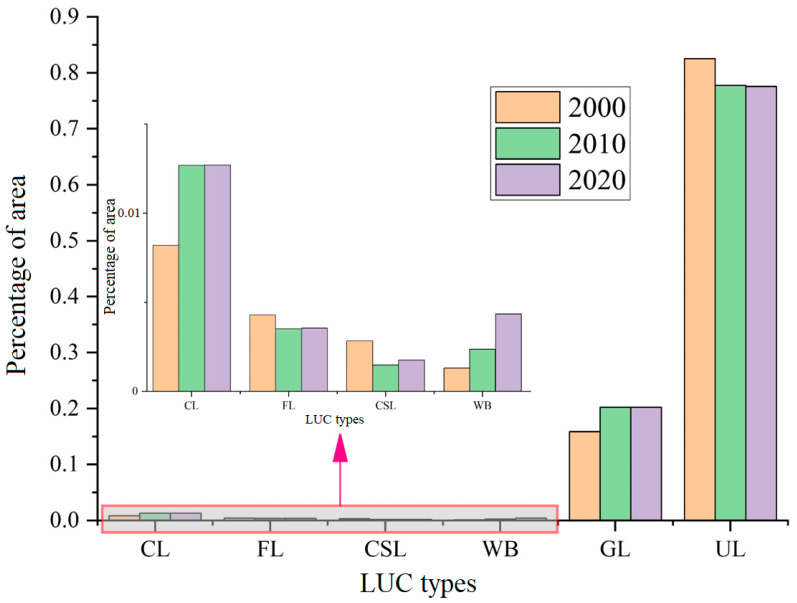
The proportions of each LUC type to area in Hami city. CL, FL, CSL, WB, GL and UL are cropland, forest land, construction land, water body, grassland and unused land, respectively.

**Figure 3 ijerph-20-04211-f003:**
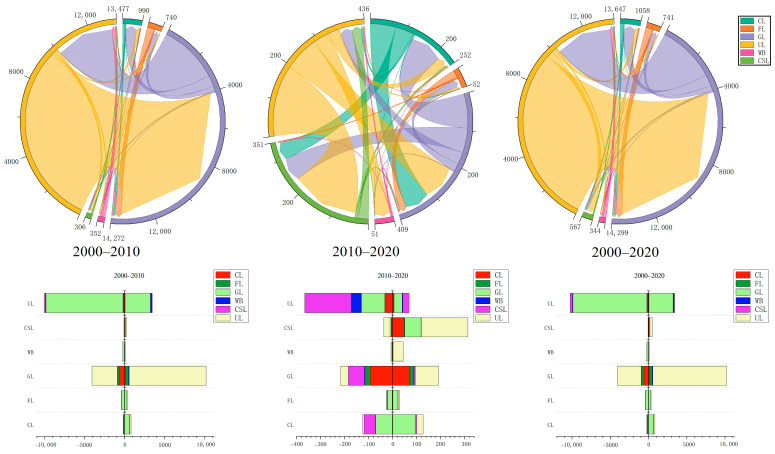
The transition among each LUC type. CL, FL, CSL, WB, GL and UL are cropland, forest land, construction land, water body, grassland and unused land, respectively.

**Figure 4 ijerph-20-04211-f004:**
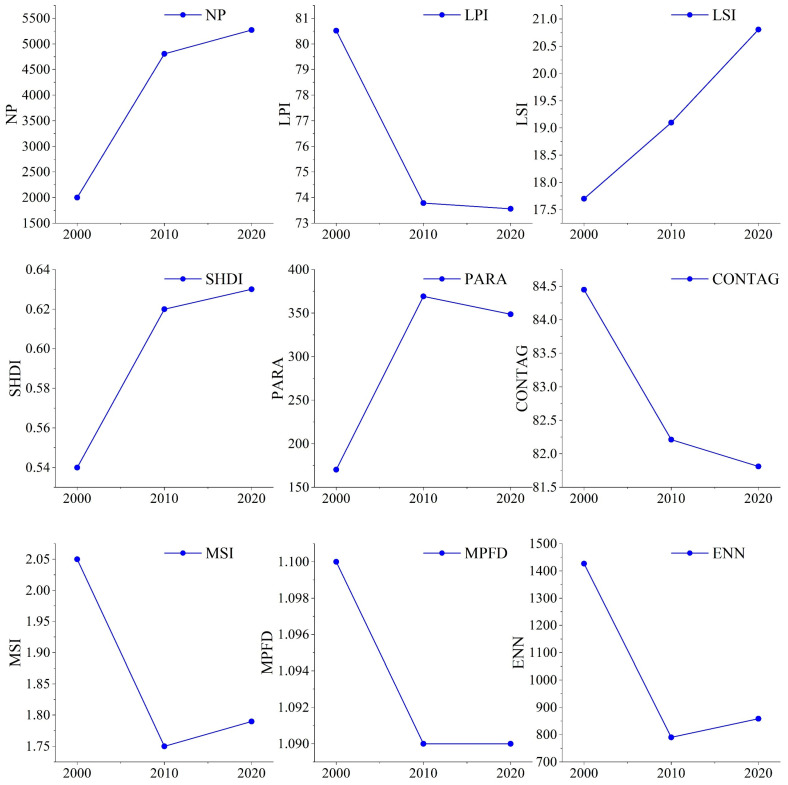
The variation of landscape metrics of the entire study area. NP, LPI, LSI, SHDI, PARA, CONTAG, MSI MPFD and ENN denotes the number of patches, large patch index, landscape shape index, Shannon diversity index, perimeter area ratio, contagion, mean shape index, fractal dimension and Euclidean nearest neighbor, respectively.

**Figure 5 ijerph-20-04211-f005:**
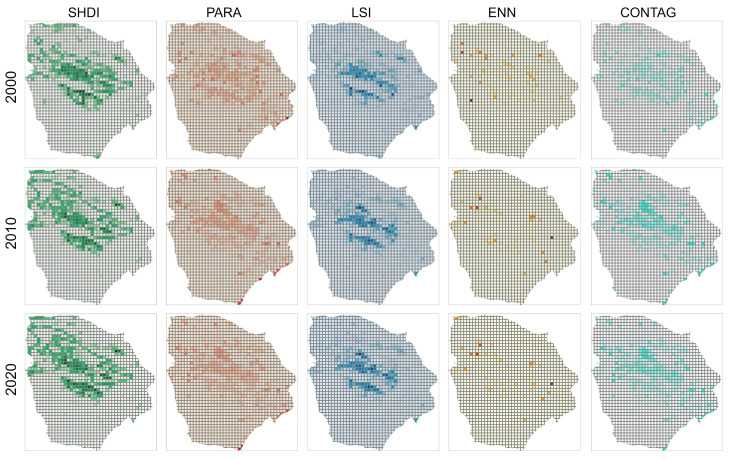
Spatial pattern of landscape metrics in the 10 km grid. SHDI, PARA, LSI, ENN and CONTAG are the Shannon diversity index, perimeter area ratio, landscape shape index, Euclidean nearest neighbor and contagion, respectively.

**Figure 6 ijerph-20-04211-f006:**
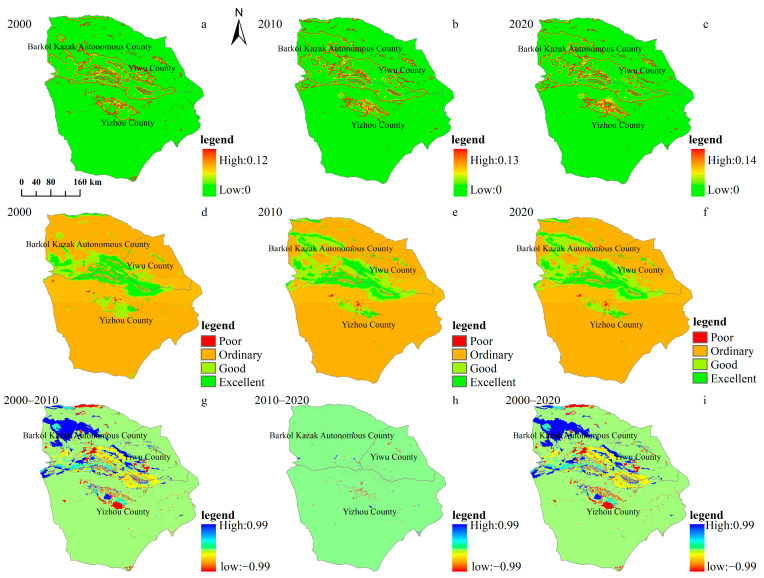
Habitat degradation degree, classes and changes in Hami city from 2000 to 2020: (**a**–**c**) habitat degradation, (**d**–**f**) habitat classes, (**g**–**i**), habitat changes.

**Figure 7 ijerph-20-04211-f007:**
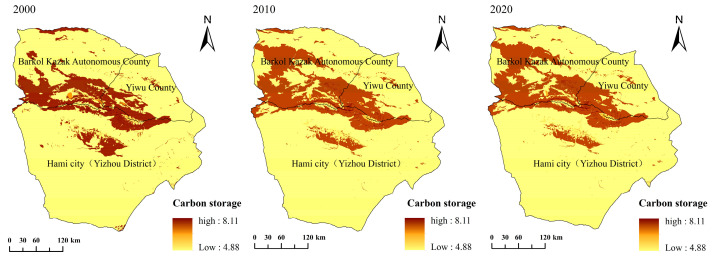
Spatial distribution of the carbon storage in Hami city.

**Figure 8 ijerph-20-04211-f008:**
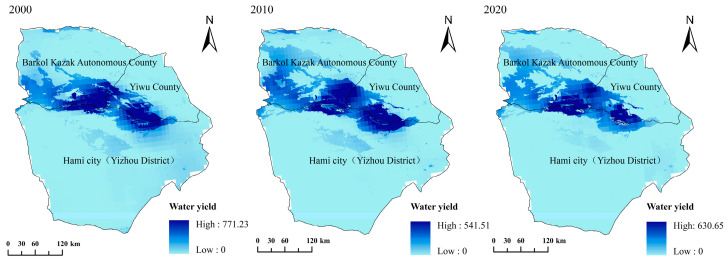
Spatial distribution of the water yield in Hami city.

**Figure 9 ijerph-20-04211-f009:**
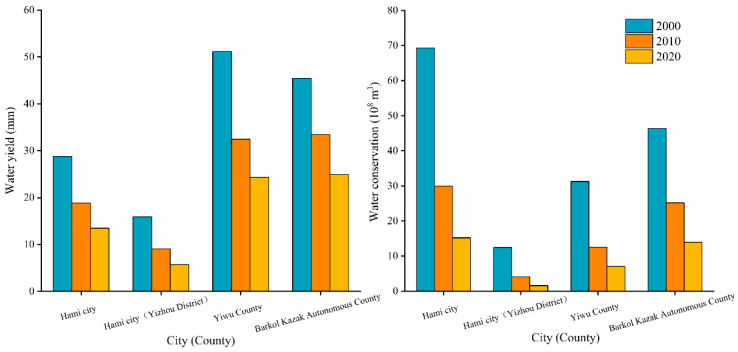
Average water yield and total water conservation by county in Hami city, 2000–2020.

**Table 1 ijerph-20-04211-t001:** Data source and parameter processing details.

Data Type	Data Sources
Precipitation	Climatic Research Unit gridded Time Series 4.04 (http://data.ceda.ac.uk/badc/cru/data/cru_ts/) (accessed on 1 January 2020), Global 0.5° × 0.5° spatially resolved annual precipitation dataset (1975–2020)
Evaporation	ERA5 (https://www.ecmwf.int/en/research/climate-reanalysis) (accessed on 1 January 2020), global 0.1° × 0.1° spatially resolved annual evapotranspiration dataset (2000–2020)
LUC	Land use data in 2000, 2010 and 2020 provided by the Land Change and Ecological Simulation Team of Xinjiang Institute of Ecology and Geography, Chinese Academy of Sciences
Soil depth	Harmonized World Soil Database version 1.2 (HWSD)
Root depth	Assignment based on vegetation type, concerning relevant studies [35]
Plant available water content (PAWC)	Based on Harmonized World Soil Database Version 1.2 at a scale of 1:1,000,000 soil data, PAWC is the difference between field water holding capacity and permanent wilting coefficient, both calculated by empirical equations; the required data include the percentage of sand, silt, clay and organic matter in the soil [36].
Runoff coefficient	InVEST user’s guide [37]
Topographic index	Calculated from basin soil depth, percentage slope and drainage area
Soil saturation conductivity	Calculated by the Neuro Theta model based on soil clay, silt and a coarse sand mass fraction [38].

**Table 2 ijerph-20-04211-t002:** Landscape metrics used in this study.

Landscape Metrics	Algorithm	Description
Number of Patches (NP)	NP=N	The total number of patches in the landscape
Largest Patch Index (LPI)	$LPI=\frac{\overset{n}{\underset{j=1}{\max(a_{ij})}}}{A}(100)$	The percentage of the landscape covered by the corresponding largest patch of each class *i*. It is a simple measure of dominance.
Landscape Shape Index (LSI)	$LSI=\frac{0.25\sum_{k=1}^{m}e_{ik}^{”}}{\sqrt{A}}$	The ratio between the actual edge length of the targeted impervious cover class and the hypothetical minimum edge length of that class.
Shannon Diversity Index (SHDI)	$SHDI=-\overset{m}{\underset{i=1}{\sum }}\left(p_{i}-\ln p_{i}\right)$	Based on Claude Shannon’s formula for entropy and estimates in the landscape diversity. The index takes into account the number of landscapes and their relative abundance (evenness).
Perimeter Area Ratio (PARA)	$PARA=p_{ij}/a_{ij}$	The ratio of the area to the perimeter.
Contagion (CONTAG)	$\left[1+\frac{\sum_{i=1}^{m}\sum_{j=1}^{n}\left(Pi\right)\frac{gik}{\sum_{k=1}^{m}gik}\times \ln(Pi)\frac{gik}{\sum_{k=1}^{m}gik}}{2\ln(m)}\right]\times 100$	The index is computed from the frequencies by which different pairs of landscape types occur as adjacent pixels on a map.
Mean Shape Index (MSI)	$MSI=\frac{\sum_{i=1}^{m}\sum_{j=1}^{n}\left(\frac{P_{ij}}{2\sqrt{\pi -a_{ij}}}\right)}{N}$	The ratio between the actual perimeter of the patch and the hypothetical minimum perimeter of the patch. The minimum perimeter equals the perimeter if the patch would be maximally compact.
Fractal Dimension_mn (MPFD)	$MPFD=\frac{\sum_{i=1}^{m}\sum_{j=1}^{n}\left(\frac{2\ln(0.25P_{ij})}{\ln a_{ij}}\right)}{N}$	Proportionate to the degree of space filled by an object withing its embedding space, is generally used to describe the fractal geometry of natural landscapes.
Euclidean Nearest Neighbor (ENN)	$ENN=\sum (ENN\left[patch_{ij}\right])$	Euclidean nearest-neighbor distance of each patch.

**Table 3 ijerph-20-04211-t003:** Threat factors regarding habitat quality.

Threat Sources	Max Distance	Weight	Spatial Decay Function
CL	4	0.6	Linear
UCL	8	0.8	Exponential
RRL	4	0.6	Exponential
OCL	6	0.7	Exponential
UL	4	0.4	Linear

Note: CL, UCL, RRL, OCL and UL represent cropland, urban construction land, rural land, other construction land and unused land, respectively.

**Table 4 ijerph-20-04211-t004:** Habitat suitability and sensitivity parameters for each LUC type.

LUC Types	Threat Source					Habitat Quality
	A	B	C	D	E	
CL	0	0.8	0.6	0.6	0.6	0.3
FL	0.7	0.9	0.8	0.8	0.8	1
GL	0.6	0.7	0.7	0.7	0.8	0.9
WB	0.4	0.7	0.7	0.7	0.7	0.9
CSL	0	0	0	0	0	0
UL	0.1	0.6	0.3	0.5	0	0.3

Note: CL, FL, GL, WB, CSL and UL represent cropland, forest, grass land, waterbody, construction land and unused land, respectively.

**Table 5 ijerph-20-04211-t005:** Carbon density for different LUC types.

LUC Types	$C_{i\_above}$	$C_{i\_below}$	$C_{i\_soil}$	$C_{i\_dead}$	Lucode
CL	3.29	0.59	50.58	0	10
FL	17.05	5.9	67.19	0	20
GL	0.4	3.97	65.8	0	30
WB	0.04	0	63.99	0	40
CSL	0.01	0	57.62	0	50
UL	0.05	0.047	54.14	0	60

**Table 6 ijerph-20-04211-t006:** The biophysical coefficients for different LUC types.

LUC Types	Evaporation Coefficient	Root Depth	Flow rate Coefficient	Lucode
CL	0.75	300	1	10
FL	0.8	3000	1	20
GL	0.65	500	1	30
WB	0.1	1	0	40
CSL	1	1	0	50
UL	1	10	0	60

**Table 7 ijerph-20-04211-t007:** Variations of landscape metrics for different LUC types.

CLASS	Year	NP	LSI	PARA	ENN
CL	2000	344	35.40	139.43	966.05
	2010	514	37.72	189.69	796.69
	2020	516	40.01	182.10	912.91
FL	2000	625	50.82	160.15	947.70
	2010	3006	75.88	448.24	299.03
	2020	2955	78.69	435.27	315.26
GL	2000	519	35.81	186.27	1577.86
	2010	643	33.30	296.73	1331.82
	2020	790	34.92	349.79	1204.37
WB	2000	78	13.88	114.67	3400.94
	2010	196	22.32	225.54	3174.51
	2020	223	22.65	215.86	3489.89
CSL	2000	268	18.22	175.83	2750.11
	2010	294	21.96	144.19	3072.55
	2020	644	34.08	146.51	2026.65
UL	2000	168	12.92	238.23	661.01
	2010	155	12.57	338.11	699.66
	2020	144	13.69	274.85	619.21

## Data Availability

The original contributions presented in the study are included in the article; further inquiries can be directed to the corresponding authors.

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
