# Peer review of "Landscape Pattern Change and Ecological Effect in a Typical Mountain–Oasis–Desert Region in the Northwest Region of China"

_ijerph, 2023, doi:10.3390/ijerph20054211_

Round 1
Author Response
Dear reviewer:
Hello!
According to your questions and suggestions, we have corrected them one by one and uploaded them in the form of Word. Please check!
Haiyan Meng
2023/2/6

Reviewer 2 Report
Review of the manuscript entitled "Landscape pattern change and ecological effect in a typical mountain–oasis–desert region in northwest region of China"
Dear Authors,
the manuscript presented for review aims to detect changes in land cover and land use as well as landscape changes in the region of Hami city in the past 20 years as well as identify the main characteristics of habitat changes in this region. The stated aim should be considered as correct. However, so that the implementation of the aim set in this way is not limited only to the study of the existing state and only in relation to the selected region, I suggest justifying the aim of the study in terms of scientific significance, i.e. I am asking the Authors to indicate what importance for the development of science and discipline and what impact on the community the results obtained in the research will be of international importance. Below are my other major comments, which I hope will assist the Authors in improving their manuscript.
1. Figure 1 shows Location of Hami city and digital elevation map as well as land use and land cover maps. However, the markings introduced in the legend have not been explained by the Authors. Part of the figure is completely illegible and not properly explained in the description - it is probably a graph of annual temperatures in the region. Thus, the caption of the figure is confusing for the reader.
2. I am asking the Authors to pay attention to the correct nomenclature related to land use and land cover. In the introduction, the Authors indicate the differences between land use and land cover, while in the further part they use these terms interchangeably or jointly. The term Land use type or the name of subchapter 2.2.2 Land use cover change also appears. Please pay attention to the differences in nomenclature and systematize this issue, in particular in subchapter 2.2.2.
3. Table 1 summarizes the data used. Each type of data has a different resolution. The Authors, however, earlier state that the resolution of the raster is 30mx30m. So are these correct? How did the Authors deal with the different resolution of individual data? How did differences in resolution affect the accuracy of the results obtained? With most data, there is also no information in what format/units/scale the data are made available and thus what modifications they have undergone in order to use them in research. In addition, it was stated that all data were expressed in the same coordinate system, i.e. CGCS 2000. I would like to ask the Authors to explain what this frame is, because for a foreign reader this designation will be unclear. I also suggest adding information from which systems the transformation took place to the CGCS 2000 system.
4. Table 2 contains a number of indicators used by the Authors. Are all these indicators relevant to the purpose of the work? Some of the indicators complement or are part of others. The list of coefficients also lacks an explanation of what these indicators depend on, there are no explained symbols and some of the same components have different symbols.
5. Line 233: no explanation of what Kx, ET0 means.
6. Figure 2: a fragment of a larger graph, it has been enlarged so that it is legible, but unfortunately the description of the y-axis is unclear and illegible (it looks like the data is in the range from 0 to 1%, which is not the case). Here again there are abbreviations for land use and land cover designations that have not yet been clarified.
7. Figure 3 is of poor quality and illegible. The upper overflow charts have a specific characteristic of complementing each other up to 360 degrees, thus what is the issue of their scale, i.e. whether the changes that took place between 2000-2010 were actually so much greater than in the years 2010-2020 and thus so significant over the years 2000 -2020? Failure to pay attention to and explain the scale of these charts may lead to erroneous conclusions and mislead the reader.
8. Figure 5 has no legend. Figures 7 and 8 have poor resolution and descriptions are illegible.
9. The work contains only a discussion, there are no separate conclusions from the conducted research. It also seems to me that 3 of the objectives of the work set out in the introduction have not been fully achieved. I suggest adding appropriate suggestions that will result from the conducted research.
There are also many abbreviations in the manuscript that have not been explained by the Authors.
Thank you.
Author Response

(The authors gave the same response as above.)

Round 2
Reviewer 1 Report
The author has made positive and effective revisions to the review opinions. I consider it has reached journal publication level and agree to accept.
Author Response
Thank you to the reviewers for their comments and responses!
Haiyan Meng
2023/2/13
Reviewer 2 Report
Dear Authors,
here are some more minor comments.
1. Tab. 2: should there be LUC types?
2. Tab. 3: Please be consistent with the abbreviations entered in the text above (CL, CSL, UL). Furthermore, Rural residental land is the same as Rural Land in Table 3? Maybe adding a abbreviation would be a good idea? Then in Table 4 A, B, C, D and E can be replaced by introduced abberviations.
3. Section 3.1: shouldn't the title be changed?
4. Figures 3 and 9: poor-quality. Also, the temperature graph in Fig. 1 is still illegible.
Thank you.
Author Response
We thank the reviewers for their comments, which will be responded to individually and uploaded in Word as an attachment.
Haiyan Meng
2023/2/13
